# Multivalent Ions as Reactive Crosslinkers for Biopolymers—A Review

**DOI:** 10.3390/molecules25081840

**Published:** 2020-04-16

**Authors:** Florian Wurm, Barbara Rietzler, Tung Pham, Thomas Bechtold

**Affiliations:** 1Research Institute of Textile Chemistry and Textile Physics, University of Innsbruck, Rundfunkplatz 4, 6850 Dornbirn, Vorarlberg, Austria; tung.pham@uibk.ac.at (T.P.); thomas.bechtold@uibk.ac.at (T.B.); 2KTH Royal Institute of Technology, School of Engineering Sciences in Chemistry, Biotechnology and Health (CBH), Fibre and Polymer Technology/WWSC, Teknikringen 56, SE-10044 Stockholm, Sweden; rietzler@kth.se

**Keywords:** multivalent ions, bivalent ions, biopolymers, crosslinking, complexation, polyamino acids, glycoproteins, interface modification

## Abstract

Many biopolymers exhibit a strong complexing ability for multivalent ions. Often such ions form ionic bridges between the polymer chains. This leads to the formation of ionic cross linked networks and supermolecular structures, thus promoting the modification of the behavior of solid and gel polymer networks. Sorption of biopolymers on fiber surfaces and interfaces increases substantially in the case of multivalent ions, e.g., calcium being available for ionic crosslinking. Through controlled adsorption and ionic crosslinking surface modification of textile fibers with biopolymers can be achieved, thus altering the characteristics at the interface between fiber and surrounding matrices. A brief introduction on the differences deriving from the biopolymers, as their interaction with other compounds, is given. Functional models are presented and specified by several examples from previous and recent studies. The relevance of ionic crosslinks in biopolymers is discussed by means of selected examples of wider use.

## 1. Introduction

Biopolymers, with consideration of their functional groups, interact with multivalent ions due to electrostatic interaction. This basic principle has been reported, and subsequently applied, in a multitude of phenomena and applications. Pectins are important structural molecules in plant cell walls which aggregate and structure in the presence of calcium ions [1], a phenomena used for decades in jam processing [2,3]. Similarly, cross-linking was reported in the cell walls of specific brown algae from the *Fucales* order. Besides cellulose microfibrils and sulfated polysaccharides, alginates structure with present calcium ions and phenols [4]. Interaction of ions and pigments in printing pastes with alginates has also been used for viscosity adjustment [5].

Ion interactions are also exploited in cellulose processing, both for their dissolution and processing [6], as their modification [7]. In one approach, banana and orange peels have been proposed to extract heavy metals from water. Due to their complexation capability these cellulose-based materials have been found promising for water purification from e.g., copper, cobalt, and zinc at trace level concentrations [8]. Complexation is also the basic principle for the production of ‘casein’ fibers [9]. In a more recent approach to enhance elastomer performance, a special type of polyisoprene crosslinking has been utilized. Ionic moieties on the polymers can hop continually between ionic aggregates, thus resulting in elastomer behavior [10].

Naturally, ion complexation is also used by silkworms for silk fibroin aggregation [11]. Also for the dissolution of these, ionic solutions of calcium chloride in ethanol (EtOH) can be applied [12].

From these few examples the scope and universality of ion biopolymer interactions can be outlined. However, electrostatic interactions are not limited to multivalent ions and biopolymeric functional groups but obviously extend to polar, polar as other charged molecules, ions, and polymers. For example, one of the basic principles of life is the iron(II)-complex in the heme of the hemo- and myoglobin, which provides human blood oxygen supply [13]. Nevertheless, for the sake of conciseness, the review is limited to multivalent ions with biopolymers, with a focus on polysaccharide based systems.

The mechanisms behind these interactions are manifold. The basic principle depends on stoichiometric considerations, ion type, and valence as present polymer functional groups. Confined to the electrostatic effect, this basic principle can affect different scales. In the absence of biopolymers, electrostatic interaction results in the formation of a hydration shell of the ion or ionic compound in solution due to the presence of polar or polarisable solvent molecules. In the presence of possible ligands, the formation of complexes is possible. Complex formation results in bridging, if ligands, or more specifically their functional groups, are part of different oligo- or polymers. Amplification of bridging, as decoiling of the biopolymers, can result in extensive viscosity increases of solutions, which finally gel upon ligand percolation. Opposite effects can include the coiling of polymers in solvent, due to reduced charge repulsion, hydrogen-bond separation, and thereby solubilization. Here the ionic biopolymer interaction influences the macrostructures of mixtures. Cross-linking can also results in coagulation of the biopolymers. In the presence of an interface, a surface, fiber, or something alike, biopolymers can undergo fixation. Similar to the scale extent of interactions of multivalent ions with ligands the manuscript is arranged.

We present common scientific understanding of interactions of multivalent ions with polysaccharides and finally take a brief look to polyamino acids. Representative interaction and crosslinking systems are presented and discussed.

After introducing hydrate shells of dissolved ions as ion complexes with simple sugar compounds, we focus on single metal ion complexes. We focus on a gluconate system, which serves as a basis for the succeeding cellulose interaction models. From these we extend our considerations to ion–polysaccharide interactions. Cellulose interactions are reviewed as cellulose interactions have been investigated in detail. Possible ionic solvents are shortly discussed. These interactions enable bridging and crosslinking of biopolymers and we pursue discussing polysaccharide-ion cross-linked biopolymer systems. Solution based crosslinking is discussed for alginate, carrageenan, and pectin systems. Specific polymers were chosen for their unique ionic structuring patterns. Besides, adsorption of alginates and pectins on cellulose surfaces are reported subsequently. This is done to extend the interaction model to ionic surface sorption of biopolymers on cellulose. Finally, to extend the models and represent them in a wider scope, we discuss the interaction of multivalent ions with polyamino acid structures. These are used as a side-glance to glycoproteins, where ionic interaction is present in the polysaccharide as the polyamino acid parts of the molecules. We present ionic interaction using polyamide as the model system. Similar considerations, with respect to solvation and structuring, to cellulose are presented. We then take a final look at silk fibroin and wool protein interactions with multivalent ions to demonstrate the general principle of ionic crosslinking on other biobased macromolecules.

## 2. Ion Hydration and Monosaccharide Complexation

### 2.1. The Hydrate Shell of Ions in Water

Interactions between biopolymers and solvated ions can lead to the formation of coordination compounds. The biopolymer takes the role of the ligand and the multivalent ion represents the center of the complex.

As the starting point for a structuring through ionic interactions between a biopolymer and an ion present in aqueous solution we have to consider the structure of the hydrated ion in solution and the ability of the biopolymer to take the function of a dissolved or insoluble ligand.

In aqueous systems, ions are surrounded by a hydrate shell, in which ligand exchange occurs. In the case of Ca^2+^ ions, the first hydration layer is formed by six water molecules (see Figure 1), which exchange rapidly, the second hydration layer contains on average 12 water molecules [14].

In aqueous solution, ions are surrounded by a hydrate shell. In this case, other dissolved or solid ligands compete for complexation, the hydrate shell is replaced by ligands which are able to form more stable complexes. The low rate of ligand exchange then permits identification of the complexes formed.

An important example for a stable Ca^2+^ complex is the formation of a stable Ca^2+^–Citrate^3-^ ion pair complex [15]. Highly stable Ca^2+^ complexes also are formed with L-tartaric acid, where the formation of [Ca^2+^Tar^2−^H_-1_]^−^ and [Ca^2+^Tar^2−^H_-2_]^2−^ (Tar^2−^ = tartrate) have been reported in alkaline aqueous solution. In these complexes, the involvement of the hydroxyl groups present in tartrate in the complex formation is discussed as one possible complex structure, another possible structure would consider Ca^2+^-hydroxo-tartrato complexes [16].

Similarly a coordination number of 4 has been reported for Mg^2+^ in aqueous solutions, where the ion [Mg(H2O)_4_]^2+^ is present, while a coordination number of 6 has been found for the primary solvation sphere of Mg^2+^ in methanol, thus corresponding to [Mg(CH_3_OH)_6_] [17].

Fe^3+^ ions are present in form of defined hexaquo complexes [Fe(H2O)_6_]^3+^ only in highly acidic solution with pH below 1, and tends to hydrolyze and condensate with formation of binuclear or multinuclear hydroxo-complexes. In the presence of stronger ligands, defined complexes with high stability are formed, e.g., the hexacyanoferrate complex [Fe(CN)_6_)]^3-^ or the trisoxalato complex [Fe(C_2_O_4_)_3_]^3-^ [18].

The selective interactions based on ion-pair formation or complex formation form the chemical basis for metal ion binding, crosslinking between polymers and formation of 3D-structures.

### 2.2. Complex Stability and Formation of Constants

For the formation of complexes in aqueous systems the concept of formation constants is a useful tool to understand and describe the stability of a complex species formed [19]. Dependent on the concentration of the center ion and ligand, as well as on pH, temperature, and ionic strength, different species of a metal–ligand system can be identified.

A representative example for the formation of different species as a function of pH and metal/ligand ratio has been shown for Fe^3+^-complexes with ß-d-gluconate [20]. In the presence of Ca^2+^-ions, the formation of mixed complexes with both, Ca^2+^ and Fe^3+^ ions, has been reported.

The example of the rather “simple” sugar acid ß-d-gluconic acid demonstrates the variability in complexes and interactions between a center ion and a polyhydroxy-carbonic acid. Often a dynamic equilibrium between two, three, or more species is present in solution. Reactions 1–7 (Figure 2) describe the formation of different complex species as the competing hydrolysis and hydroxide formation (reaction 7). At a ligand to metal ratio of 1, mainly 1:1 complexes between Fe^3+^ and D-gluconate (DGL) are formed, in presence of Ca^2+^ 1:1:1 complexes of Ca^2+^/Fe^3+^/DGL are present. At a ligand to Fe^3+^ ratio of 2 mainly 1:2 complexes for Fe^3+^-DGL and 1:1:2 complexes for Ca^2+^/Fe^3+^/DGL are formed (Figure 2).

The model of iron–sugar acid complexes demonstrates two major principles which are of importance to understand the formation of ionic crosslinks between polysaccharide structures:In analogy to polyhydroxycarboxylic acids also polysaccharides will be able to form a high number of complexes with multivalent ions, e.g., calcium and iron.The formation of complex species with 1:1:1 stoichiometry (e.g., [CaFeIIIH-3DGL]^+^ at pH 7) indicates that the presence of a carboxylic group may support formation of stable complexes at low pH, however participation of hydroxyl groups will also be contributing to complex formation. As an example, in highly alkaline aqueous solution the formation of iron-complexes with sorbitol complexes can be observed.

Polysaccharides in solid state as dissolving in water, thus, can be understood as multi-dentate ligands, which can form a high number of complexes with multi-valent ions. In solid polysaccharide structures, the amorphous parts of the material will be accessible for dissolved ions, where complexation of metal ions occurs accordingly. The complex formation with a polysaccharide is in competition with the formation of soluble complexes in water, e.g., aquo-complexes, as complexes with dissolved ligands and with the presence of a rapidly exchanging hydrate shell.

Different reactions will be observed dependent on the type of polysaccharide, the competing complex species in solution and the composition of solution used:Adsorption/complexation of metal ions e.g., Ca^2+^ and Fe^3+^ from aqueous solution into a solid polysaccharide matrix occurs,Dissolution of polysaccharides into the concentrated metal complex solutions can be achieved, e.g., dissolution of cellulose into alkaline Fe^3+^ tartaric complexes, andGel-formation and precipitation of dissolved polysaccharides, e.g., alginates in presence of multi-valent ions (Ca^2+^), occurs due to formation of metal complexes with reduce solubility. As a result a polymer network is formed.



**Stoichiometry Ca^2+^:Fe^3+^:DGL 1:1:1  increasing pH**
Ca^2+^ + Fe^3+^ + DGL↔[CaFe^III^DGL]^4+^ 


 (1)Ca^2+^ + Fe^3+^ + DGL↔[CaFe^III^H_-1_DGL]^3+^ + H^+^ (2)Ca^2+^ + Fe^3+^ + DGL↔[CaFe^III^H_-2_DGL]^2+^ + 2H^+^ (3)Ca^2+^ + Fe^3+^ + DGL↔[CaFe^III^H_-3_DGL]^+^ + 3H^+^ (4)Ca^2+^ + Fe^3+^ + DGL↔[CaFe^III^H_-4_DGL] + 4H^+^ (5)Ca^2+^ + Fe^3+^ + DGL↔[CaFe^III^H_-5_DGL]^−^ + 5H^+^ (6)


Competing reaction
Fe^3+^ + 3 H_2_O↔Fe(OH)_3_ + 3H^+^(7)

## 3. Metal Ion Interactions with Polysaccharide Structures

### 3.1. Metal Complex as Structure Model

Metal complexes with hydroxycarboxylic acids can serve as models to understand the interaction of metal ions with polysaccharide structures. For example, citrate acts as a tridentate ligand with involvement of two carboxylic groups as the hydroxyl group (Figure 3) [21]. The formation of [CaCitr]^−^ complexes has been reported for solutions with pH above 4 [22,23].

To side-glance, this complexation of calcium ions by formation of chelate complexes with citrate is used to remove calcium bound in caseins, thus modifying the texture and increasing the thermal stability of milk while improving the processability of melting cheese [24].

Another example is tartaric acid which is able to form stable iron(III)-complexes in alkaline solution. Dependent on the composition of the solution two main species are formed [(C_4_H_2_O_6_)Fe]Na at a molar ratio of Fe^3+^:tartaric acid:NaOH of 1:1:1 and [(C_4_H_3_O_6_)_3_Fe]Na_6_ at a molar ratio of 1:3:6, respectively [25]. A proposed structure for the complex between iron(III) and tartaric acid at a molar ratio of 1:3 [(C_4_H_3_O_6_)_3_Fe]Na_6_ is shown in Figure 4. In concentrated solutions, these complexes dissolve cellulose, the complexes with highest dissolution power were found with a molar ratio between iron and tartaric acid of 1:3.

In a similar manner, stable Fe^3+^ complexes have been reported to form with galactaric acid, and D-glucosaminic acid. Low complex stability was reported for D-glucosamine, indicating the importance of a carboxylic group for Fe^3+^ complexation [27].

Similar complexes with organic acids, e.g., oxalic acid, lactic acid, or citric acid did not lead to dissolution of cellulose [28]. A major reason for this finding results from the lower formation constant of the iron complexes, which hydrolyse in alkaline solution with precipitation of iron hydroxides. Thus, no stable complexes for exchange of the ligand are formed in the highly alkaline solution, which however would be required to achieve cellulose dissolution through formation of cellulose–iron complexes.

### 3.2. Ion-Uptake in Cellulose

In the cellulose molecule a high number of functional groups are present which can contribute to the binding of metal ions. Each anhydroglucose unit bears in total three hydroxyl groups, one each at carbon atom C-2, C-3, and C-6. The polymer chain is formed by a glycosidic linkage between the C-1 and C-4 of two glucopyranosyl units. Thus, at one end of the cellulose chain a free hydroxyl group is present at a C-4 atom. This end of the polymer chain is called the non-reducing end, while at the other end of the polymer an aldehyde groups is present, which easily becomes oxidized to the corresponding carboxylic group (Figure 5). This so-called “reducing end” of the cellulose chain becomes oxidized during purification and chemical processing of cellulose. Thus, dependent on the chain length of the cellulose polymer a stoichiometric amount of carboxylic groups is present in processed celluloses.

Representative values for cellulose fibers are in the dimension of 15–20 mM carboxyl groups per kg of fibers. (Viscose fibers: 20 mM/kg, modal fibers 17 mM/kg, and lyocell type fibers 15–16 mM/kg) [29].

Due to other oxidative reactions also carboxylic groups can be formed at the C-6, e.g., by selective oxidation with TEMPO (tetramethyl-piperidine-*N*-oxide) [30], or at C-3 and C-4. This, however, is accompanied with destruction of the pyranoside ring and reduction in degree of polymerisation.

The accessible carboxylic groups and hydroxylic groups in swollen cellulose form a structure, which is very similar to the dissolved hydroxycarboxylic acids previously discussed. Thus, cellulose exhibits a high number of potential binding sites for binding of multivalent ions through ion-exchange and complex formation. The uptake of Ca^2+^ ions in a cellulose structure follows a distinct stoichiometric reaction. Thus, a saturation point is reached at a concentration where all accessible carboxylic groups have been transferred into the corresponding Ca^2+^-salt (Figure 6).

The binding reaction corresponds to Equation (8):(8)$$Cellulose-COOH+Ca^{2+}+anion^{-}\to Cellulose-COO^{-}Ca^{2+}anion^{-}+H^{+}.$$

The stoichiometry between carboxylic groups present and Ca^2+^ ions bound, stabilizes at a ratio of 1:1. As the number of carboxylic groups in the cellulose structure is rather low the geometric distance to a next carboxylic group is too far to achieve a 2:1 salt with the stoichiometry of (Cellulose-COO)_2_Ca.

A similar behavior is observed in adsorption/complex formation with other metal ions, e.g., copper, iron, and zinc in cellulose. Accordingly, the binding of Fe^3+^ into the cellulose structure follows a 1:1 stoichiometry between the carboxylic groups available and bound Fe^3+^ [32]. In the case of Cu^2+^ in alkaline solution, the amount of bound Cu^2+^ exceeds the number of carboxylic groups by far, thus indicating a complex formation similar to the Cu^2+^ sorbitol complex system [33].

The selective uptake of multivalent ions, e.g., Ca^2+^ or Fe^3+^ from aqueous phase into a swollen cellulose structure also forms the chemical basis for further attachment of polysaccharides and other biopolymers to the surface of cellulose. The adsorbed multivalent ions builds a charged site, which can act as an anchor for binding of polysaccharides or other charged molecules from solution.

### 3.3. Cellulose Solvents

In this paragraph, the dissolution of cellulose will be taken as a representative example for the interaction of metal ions with a polysaccharide. Dissolution of cellulose can be achieved through different principles:by derivatisation of the functional groups (hydroxyl groups C-2, C-3, and C-6) e.g., through xanthogenation, acetylation, or alkylation,formation of metal complexes, e.g., with iron-tartaric acid, copper-amino complexes, or theuse of cellulose solvents, e.g., NMMO (*N*-methyl-morpholine-*N*-oxide) and ionic liquids.

The formation of soluble metal complexes between cellulose and an appropriate concentrated aqueous metal complex can be regarded as a model for metal-complex-based bridging between macromolecules, which share a common center ion. The polyhydroxy compound cellulose acts as a polymer ligand, offering a high number of binding sites and thus forming a high number of complexes along the polymer chain with the metal ions present in the solvent. During dissolution of cellulose, both, amorphous and crystalline domains of the polymer, dissolve, therefore in solution these complexes are formed along the full cellulose chain.

The dissolution of cellulose cannot be explained by involvement of the few carboxylic groups present at the end of the cellulose chain at the reducing end, but also requires complex formation with the hydroxyl groups on C-2, C-3, and C-6. The anhydroglucose units participate in the complex to a major extent with their hydroxyl groups. According to the literature, mainly the glycol moiety C-2 and C-3 are responsible for alkaline iron–tartaric acid complexation [34]. This is also the case for cuprammonium [Cu(NH_3_)_4_](OH)_2_ cellulose solvent systems.

Figure 7 shows an example of the proposed mixed complex between the cuprammonium system [Cu(NH_3_)_4_](OH)_2_ and cellulose. Figure 8 depicts a proposed structure for the complex formation between the alkaline iron(III)–tartrate complex and cellulose [35].

## 4. Metal Ion Based Cross-Linking of Polysaccharides

Crosslinking of polysaccharides occurs in case two polysaccharides share their function as a ligand in a metal complex.

From the model consideration with hydrate shells, soluble metal complexes and the analogies to cellulose solvents we can distinguish between three general principles of metal polymer interactions, which form a chemical basis for crosslinking of biopolymers at surfaces and interfaces.

Formation of ion-rich hydrate shells around biopolymers leading to dissolution of insoluble polymers, e.g., carrageenan and silk [12,37].Formation of defined complexes with involvement of functional groups which permit complexation at neutral pH, e.g., carboxyl groups and amino groups (Figure 9) [38].Formation of complexes at higher alkaline conditions with involvement of hydroxyl groups, e.g., the C-2 and C-3 groups [35].

For the formation of ion-rich hydrate shells the presence of highly concentrated solutions, e.g., CaCl_2_/ethanol/water of 8 M, urea or LiBr are required. Upon dilution, a destabilization of the solvent state leads to precipitation and regeneration of the polymer. The center ion of these weak complexes will then be released from the complexes and be washed out.

In a similar manner, complexes which base on the involvement of the glycol moiety in highly alkaline conditions will lose stability at lower pH, thus, polymer precipitation will occur upon dilution and reduction in pH. Only a minor amount of complexes which involve the carboxylic groups present in cellulose will be stable enough to exist at neutral pH.

Complexes which base on the second principle exhibit sufficient stability to form complex structures at weakly acidic, neutral, and weakly alkaline conditions. Dependent on geometric conditions, concentrations of metal ions, number of complex forming sites in the polysaccharide, solution conditions, and method of preparation the metal complexes are formed by a polymer chain alone or form a bridge between two polymer chains, which both act as ligands in a joint metal complex. In the following, complexes based on the second principle will be described in more detail. For the sake of representative system we limit the consideration to alginates, with an extension to similar pectin cross-linking, and carrageenans before a side-glance to related biopolymers.

### 4.1. Alginate

Alginates are salt forms of linear copolymers of 1,4-linked β-d-mannuronic (M) and α-l-guluronic (G) acid residues, known as alginic acid. They are extracted from different species of brown algae found at the coasts of Asia, the northern Atlantic as South America [39]. These residues are arranged in blocks where MG blocks alternate with mono-type block structures [40,41]. Composition of the copolymer is statistically distributed and shares of the monomers depend on region, season and plant age at harvesting [42]. Alginic acid itself is insoluble in water, which is not the case for its monovalent salt form. Solutions of these polysaccharides are highly viscous, dependent on concentration and presence of ionic species. Alginate molecules in solution were found to behave like extended flexible coils.

GG blocks in alginates selectively bind to calcium ions [41,43] as several other cations, including iron (III), strontium, and barium. Bivalent ions lead to inter-chain association of the polymer molecules. Associations of GG blocks with these bivalent cations was found to happen via an ‘egg-box’ structuring, stoichiometrically requiring half as many calcium ions as poly-l-guluronate dimers [43,44]. Computational simulations suggested a four-oxygen coordination [43]. Due to the configuration of the G-units in the alginate molecules, adjacent monomer units form a three-dimensional-box structure, which encapsulates these ions. Given a second box on the opposite side, a crosslink between two polymers is formed. The computational model was further enhanced, revealing the difference in parallel and antiparallel binding. Parallel chain association shows a fair electrostatic input from calcium coordination by five oxygen atoms of the guluronate atoms and weaker hydrogen bonds between O-3 and O-5’ of the adjacent chain. For the antiparallel arrangement, hydrogen bonds between O-2 and O-6’, as O-3 and O-5’, are mainly dominating the connection. The unique calcium coordination site in the antiparallel system is tetradentate with respect to the surrounding oxygen atoms [45]. The ‘egg-box’ model is therefore partly incorrect, though is still regarded as principally correct with respect to the offered intuitive understanding. A sketch of the model was added in Figure 10 and Figure 11.

The distribution of these G-acid distributions along the alginate sample therefore limits the length of the junction zones. The size of these junction zones can be estimated using SAXS [46]. Strength as selectivity of the binding depends therefore directly on the size of the bivalent ion in the ‘boxes’. Ion affinity was found to order with Ba^2+^ > Sr^2+^ > Ca^2+^ > Mg^2+^ [47,48]. Subsequent studies corroborated with the model by Atkins et al. [49] showing a structural repetition in G-unit rich alginates of 0.87 nm in the acid form [50]. The molecular structure, with and without binding cations, was also investigated in modeling simulations by Braccini et al. [51]. Results show that polyguluronic acid structures to two-fold helices, in gel as solid form, in various salt forms. By contrast, polymannuronic chains can also adopt to a three-fold helical structure in solid salt form. From the helical structure it is also shown that for Ca-guluronate the most favorable sites are periodic, identical, and tetradental chelating along the chains. This bases on the suggestion from Angyal [52] for an efficient hydroxyl binding pattern if these are in an axial–equatorial–axial configuration with respect to the guluronate residues. Moreover, Ca–guluronate binding is suggested to not be purely electrostatic, which is the case for polymannuronic chains, the latter showing no calcium specificity. A more detailed conformational calculation for mixed guluronic–mannuronic forms was not included in the study and probably is of lower significance due to non-selectivity of ions of mannuronic residues. In contrast, crosslinking of long blocks of alternating G- and M-units in alginates was experimentally found present, also leading to the formation of mixed GG/MG junction zones [53]. These ‘secondary’ MG junctions enable syneresis in gels, if a certain extension is present. Vice versa short junctions impair syneresis appearance. This finding was backed by simulation results for affine deformation theory of Gaussian networks [54].

Alginate gelation has been described by a two-stage process [45]. After dimerization of the polymers, these interact weakly through non-specific electrostatic interactions to form associates. From dilute solutions the process was shown to include a third stage, which is monocomplex formation before subsequent dimerization and associate formation [55]. These associates are suggested to form from dimerized polymers packed on a hexagonal lattice structure. This hexagonality is not a true geometrical as in crystallography, but rather a located order [50]. Lateral interaction of dimerized polymers happens via hydrogen bonding, water molecules or present sodium and calcium ions [46,50]. It was also reported that calcium coordination reduces the ability for smectic lateral ordering, which is more pronounced in the acid form and, thus, increases nematic ordering [50]. The resultant junction zones depend therefore on block length distribution of G-units, concentration of binding ions as polymer concentration. For alternating MG junctions, interaction findings were only based on the presence of calcium ions [53].

Beside physically bound alginates, chemically cross-linked alginates have been produced. Alginates can be derivatized by multiple approaches, cell receptors can be introduced onto the polymer chain and mixtures with additional polymeric species was investigated on. Partly oxidized alginate structures can also enhance biocompatibility. We point the interested reader to the reviews of Andersen et al. [56], Sun et al. [57], and Yang et al. [58].

Interaction possibility with bivalent ions, crosslinking, and gelling properties of alginates led to their use in a high number of applications. Alginates have been exploited as a functional food ingredient [59], in textile printing and dying [5], as an immobilizing agent for enzymes and cells [57,60,61], in pharmaceutical and medical uses [62,63,64], and various other applications [65]. In textile printing and food applications usually the thickening and gelling properties are used, given a sufficient amount of bivalent ions. Final adjustment in texture of the products is based on present ionic species as cost sensitivity. Usually, the share of G/M of the alginates are of minor importance. In medical and pharmaceutical uses, mostly the gelling ability of the alginates is exploited. Medical application of alginates is possible due to the lack of an immunogenic response to the biopolymer, given adequate purity of the compound [63,66,67]. One possibility of applications are pharmaceuticals and enzymes packed into alginate gel beads. Due to the nanoporous structure these can diffuse out. Alginate wound dressings, besides having a barrier function, are kept moist and enhance wound healing. Due to low protein adsorption and the lack of cell receptors, alginates are a specific model for cell culture and biomedical studies. The latter options have been exploited in tissue regeneration studies. Blood vessel formation is driven by selective introduction of cells as angiogenic proteins and genes in alginate matrices. A similar approach is the delivery of osteoinductive factors and cells in bone regeneration, chondrogenic cells in damaged cartilage as cell transplantation and growth factor in other tissues and organs [57,63]. Several trials of an injectable alginate matrix including cells have been conducted [61,64]. Studies and investigations are numerous and the above given overview is just a brief scope. The G/M composition of the alginates is of high importance in medical and pharmaceutical applications.

### 4.2. Carrageenans

Other marine polysaccharides widely used are carrageenans, which aggregate in ionic solutions very different from alginate or chitosan, but similar to agarose. First, we present general information, such as structuring, before crosslinking and ion selectivities are reported.

These extracts of various red seaweeds are collected in Asia, the northern Atlantic, and South America. These polysaccharides are a hydrophilic family of biopolymers, their backbone consisting both of sulfated and non-sulfated galactose and 3,6-anhydrogalactose (AG) monomers. The monomers are linked by alternating α-(1,3) and β-(1,4) glycosidic bonds. The number of sulfate ester groups as 3,6-anhydrogalactose units in the polymers are used to categorize the polymers into different types. Each of these types shows different properties in application due to differences in conformation and configuration [39].

Main carrageenan types are divided into ι- (iota), κ- (kappa) and λ-carrageenans (lambda) with precursor types μ- (mu) and ν- (nu), as postcursor θ-carrageenan (theta) and κ-related furcellaran (Fur), which is referred to as β/κ-hybrid, where β are non-sulfated κ-carrageenans. More different types were defined but since they are only of minor importance we did not include them. See Figure 12 for a graphic representation of the different carrageenan types.

In natural seaweed, substantial amounts of ν- and μ-carrageenans are randomly distributed in κ- and ι-carrageenan. These are modified using alkali in the extraction procedure to derive more pure κ and ι products [39]. Basic differences in ester sulfate content as 3,6-anhydrogalactose content for the different types are given in Table 1. Carrageenans never are present in a pure form [68]. Heterotypes might be present as separate chains or as distribution in the chain of the dominant type, be it randomly or in regular patterns [69]. As mentioned for alginates, chemical composition of carrageenans is also dependent on seasonal variability, age of plant, and region of harvesting [70].

ι-, κ-carrageenan and furcellaran are forming helical structures found in solid state as in solution below a certain transition temperature. This temperature is difficult to specify as carrageenan samples usually are polydisperse and dependent on ionic concentration present [69]. This is the first step in an aggregation and gelation mechanism. The latter mechanism was proposed to happen via a ‘domain model’ [73], which is sketched in Figure 13.

It is possible to form non-aggregated solutions of κ- and ι-carrageenan, though these are highly dependent on temperature, concentration, and added salt [74].

ι-carrageenan is found to structure in right-handed double helices, half-staggered, where one helix is exactly displaced half a pitch relative to the other, and parallel. The threefold helices order in a pitch of 2.6 nm [69]. For κ-carrageenan the helical structures were derived from fitting models to continuous X-ray diffraction data as direct measurements were not meaningful. Best solutions were found for a three-fold double helix with a pitch of 2.5 nm. By contrast, the double-helices of κ-carrageenan show an offset from a half-staggered position by 28° and 0.1 nm translation [75]. This latter translation in κ-carrageenan has been questioned by Cairns et al. [76]. For Fur, showing nearly the same helical structuring as κ-carrageenan, the axial translation in the double helical conformation is present though [76].

λ is a non-gelling carrageenan and does not form helices [69]. It shows considerable flexibility, no evidence for association and dissociation in solution and a non-ideal behavior in polyelectrolyte suppressing conditions, probably due to large excluded volumes [77]. Though, at high present ionic concentrations even λ -carrageenan forms gels [39]. Besides, gelation of λ-carrageenan has been reported in the presence of trivalent ions [78], which might enhance its applicability.

There is a high sensitivity of the mentioned types to present ions, which not only depends on type and valence but also on the identity of the ions present. Since interaction and crosslinking with ions is determined by the carrageenan type, we describe the different interaction mechanisms for these separately.

#### 4.2.1. κ -Carrageenan and Furcellaran

κ undergoes conformational transitions at present ion concentrations. Conformations are highly selective for a selection of monovalent ions (e.g., Cs^+^, K^+^, and Rb^+^) and rather unselective to both divalent ions (e.g., Mg^2+^, Ca^2+^, and Ba^2+^) and other monovalent ions (including Li^+^ and Na^+^). The latter showing the lowest helix-forming efficiency [69]. It seems from various studies, that these selective ions bind to the polymer even in non-aggregating conditions, whilst the non-specific including the bivalent ions, rather interact via long-range coulombic interaction. Though binding is present, no evidence of a cationic binding site could be determined. Cation specificity can be explained only by the assumption of one–two binding sites per disaccharide [79]. Several anions (including I^−^, F^−^, and Cl^−^) also have been found to stabilize helical conformation, which is suggested to happen through binding to the molecule. There seems to be strong evidence for specific anionic and cationic binding sites, but no final investigation proves these assumptions [69]. κ is mostly gelled and investigated on using monovalent ions and minor influences of divalent ions are mentioned in several studies. For the sake of concentrating on multivalent ions, we omit going into further detail.

Furcellaran shows a similar behavior with regard to present ions as κ-carrageenan. Transition temperatures are higher for furcellaran than κ-carrageenan though [80,81].

#### 4.2.2. ι-Carrageenan

In comparison to κ-, ι -carrageenan helices are strongly stabilized by divalent ions (e.g., Ca^2+^) [82], mostly due to the higher valence of the ions interaction non-specific with the higher charge density polymer [83,84]. Some studies found an influence by monovalent ions [85], which was later deduced to probably derive from κ-carrageenan impurities [86]. Apart from minor transition temperature changes, there seems to be no indication of cationic specificity for conformation transitions given a high purity ι-carrageenan [69]. We can therefore conclude that divalent ions act rather unspecific, more markedly than monovalent ions, and are frequently used in gelation of ι structures.

#### 4.2.3. Aggregation and Gelation

κ forms bigger aggregates from double helices, which resemble microfibers. The exact structure in these microfibers could not be deduced, but they were found to consist of several carrageenan helices. Microfibers then seem to form bundles and bigger aggregates [87]. Further SAXS studies revealed that microfibrilliar aggregates are polydisperse [88]. These microfibrils are a requisite for gelation, which happens further through lateral accumulation [69], but could locally also be driven by branching [89]. Yuguchi et al. [88] did not find any further aggregates for ι type systems after double helix conformation. These helices network further through ionic mediation. There seems to be a higher tendency for branching of double helices for ι-carrageenan than for κ. No aggregations details for furcellaran were reported.

The water-holding, stabilizing, and thickening capabilities have been exploited in various different applications. The main application of carrageenans is food processing [90,91]. There mostly is no single specific interaction of divalent ions with the carrageenans present, but rather a mixture of interactions, including strong interaction with dairy ingredients (e.g., casein). Therefore we do not further specify all different applications but refer the interested reader to the reviews of Campo et al. [92] and Piculell [69].

### 4.3. Pectin

Pectins are a family of polysaccharides, rich in galacturonic acids, which are present in primary and secondary cell walls of plants. The backbone of the polymer comprises of 1,4-linked α-d-galacturonic units, disturbed by 1,2 linked α-l-rhamnose residues [93]. The amount of disturbing rhamnose units is dependent on the pectin source. It is further suggested that pectin structure interchanges between rather regular galacturonic and irregular rhamnose rich regions. The former regions being smooth, while the latter are described as hairy due to neutral sugar side chains attached to the rhamnose residues or the glacturonic acids [94,95]. Side chains are mainly galactans, arabinan, or arabinogalactans, having itself single xylose side chains or galactan side chains [96]. In special pectins, minor amounts of acetyl and feruloyl groups are found [97]. Native pectins show complex structures including various sugars bound to the galacturonic backbone and different grades of branching. Industrial pectins usually are less complex due to hydrolysis breakdown in the extraction process.

The galacturonic acids’ carboxyl groups are present either in salt form or are esterified by methanol. These carboxyl groups can be ionized, leading to classify pectin as polyanion. The linear charge density of the molecules is mainly influencing the interaction between counter-ions and the polyelectrolytes. Since the charge density is dependent on the degree of esterification (DE), the esterification is the main influence determining the ion binding of pectins [97,98]. The DE also affects solubility, thickening, as well as other properties. As direct consequence, the pH and the ionic strength affect cation binding to pectins. It was found that calcium binding reaches a maximum level at pH 5– 7.5, and levels decrease if ionic strength increases [99]. Calcium binding bases on nonspecific electrostatic interaction as on coordinative binding. This cooperative binding is supposed to go along with conformational changes of the polysaccharide, and has also been proposed for pectin gelation via the egg-box model [43,100]. Since pectins are highly prone to aggregation in solution, solvent quality, degree of esterification, and charge density as neutral sugar content has to be controlled to obtain macromolecular solutions. The lack of attention on all factors might have led to quite discrepant findings in light scattering studies, as pectin data for the Mark–Houwink–Sakurada equation (see [97] for a detailed overview of values). In dilute solutions pectin chains are found stiff and extended, due to the relatively rigid galacturonic backbone. Rhamnose residues kink the pectin chains [101]. However, these kinking effects were found to partly self-eliminate themselves due to successively paired rhamnose units [102]. Other contradictory conclusions have been drawn about the influence of the DE on the pectin conformation. While several studies suggest that there is no significant effect of the DE on the conformation [103,104], others report decreasing hydrodynamic volumes [105] and increasing chain stiffness [106,107] if the DE is decreased.

Gelation usually is divided into two categories. Gels of high- and low-methoxylated pectins. The former are a complex mixture of intermolecular interactions and are usually formed in high sucrose concentrations or other co-solutes and at low pH (2.5–3.5). The low ionization level of the few carboxyl groups results in low electrostatic repulsion. Gels then form through helix aggregation, stabilized through intermolecular hydrogen bonds, as hydrophobic grouping of methyl esters [108,109]. In the case of low-methoxylated pectin gels, divalent cations, calcium being the standard example, mostly bind to accessible ionized carboxyl groups on the galacturonate units. Egg-box bonds have been proposed to occur in a two-step process, where molecular dimerization is followed by subsequent aggregation of these dimers [100]. As mentioned for alginates above, the calcium ions are suggested to occupy the electronegative center of a type of box structure that forms through clamping galacturonic acid residues on adjacent chains. Walkinshaw and Arnott [108] further speculate that calcium might bind to three oxygen functionalities on one chain and to two on the adjacent antiparallel chain. The possible binding sites for calcium ions were corroborated by Braccini et al. [51], adjusting the model in a follow-up study [45]. This study accounts for the fact that the most favourable antiparallel associations in polygalacturonate associations requires a shift (1.7 nm) of one chain compared to the other. This shifts leads to efficient van der Waals contacts, it reduces the prior large cavity to subcavities in the calcium ion size and it accounts for efficient intermolecular hydrogen bonding of the chains. The authors refer to a ‘shifted egg-box model’.

As mentioned for the compounds above, the gelling and thickening properties of pectin have been exploited in similar applications. Pectin mostly is used in food processing, including jams, soft drinks, and tart glazings [97]. Besides pharmaceutical applications it is used in edible films, paper substitutes, and many others [110]. We ask the interested reader to refer to the referenced studies.

### 4.4. Xanthan and Other Polysaccharides

Xanthan is a polysaccharide with a cellulose backbone and various possible side chains on C-3 of the glucose residues. Weak gelation is enhanced using bivalent or trivalent ions [111,112,113]. No conclusive model for the structuring, single or double helicity, as for the aggregation of the chains has been distilled out of the multiple studies performed on the structures. Due to the heterogeneity of the compounds, structural investigations are difficult and various structures might be present in parallel [114]. Studies show though that ionic interaction and linking of the polymer chains happens partly via interaction of the end groups of side chains with present bivalent ions [113]. Xanthan shows synergistic increase in viscosity if mixed with several galactomannans.

There are numerous other examples of interactions of polysaccharides with bivalent ions. For the sake of the finiteness of this part, we direct the reader to the respective books and articles therein, where further interaction systems are described [39,114].

## 5. Metal-Ion Based Crosslinking on Fiber Surfaces and Interfaces

Cellulose fibers represent a solid structure which offers binding sites for metal ions with involvement of carboxylic groups. Thus, metal ion complexation can occur at neutral pH. The complexation of positive metal ions in the structure also influences the negative zeta potential [8]. In sulfonated microcrystalline cellulose, the introduction of sulfonate groups led to an increase in negative zeta potential from -14.3 mV in neutral solution to -37.9 mV. Uptake of metal ions Fe^3+^, Pb^2+^, and Cu^2+^ reduced the negative potential to −6.24, −9.7, and 14.4 mV respectively [115].

The sorption of metal ions on the insoluble cellulose structure leads to the formation of possible binding sites to form complex bridges with dissolved polysaccharides. Accessibility of the possible binding sites is a decisive condition for the successful formation of such mixed ion-complex bridges. Due to the size of these polysaccharides, the binding and deposition of a polymer will occur only at the surface of the insoluble cellulose structure.

The surface modification of Ca^2+^-treated cellulose by sorption of pectin, alginate, and xanthan could be demonstrated using a two-stage procedure [116]. In the first step, the cellulose fiber samples (viscose or lyocell type fibers) were impregnated with CaCl_2_. Through spontaneous sorption the binding sites for later sorption of a dissolved polysaccharide were formed. In a second step, the samples were immersed into polysaccharide solutions. Due to the presence of carboxylic groups in the polymer chain of pectin, alginate, and xanthan, sorption on the surface of the calcium containing fibers occurred. The amount of sorbed polysaccharide increases from 0.02 mg/g polysaccharide on calcium untreated fibers to 0.2 mg/g polysaccharide on calcium pre-treated ones. A representative molecular structure for the principle of ionic crosslinking between carboxylic groups of cellulose (at C-6) and pectin is shown in Figure 14.

A reaction scheme can be formulated according to the Equations (9) and (10), where PS are carboxyl-containing polysaccharides:(9)$$Cellulose-COO^{-}+Ca^{2+}\to Cellulose-COO^{-}Ca^{2+}$$
(10)$$Cellulose-COO^{-}Ca^{2+}+\mathrm{PS}-COO^{-}\to Cellulose-COO^{-}Ca^{2+}COO^{-}-\mathrm{PS}.$$

In another approach, the metal complex with Fe^3+^ is formed in solution first. Then sorption of the soluble polysaccharide complex on cellulose is achieved in the second step (Figure 15) [117]. Alginate has been used as soluble ligand offering a number of sites to form metal complexes, which equilibrate. In the presence of cellulose surfaces, the sorption of the complexes to the fibers surface occurs. In this case, the swollen cellulose structure serves as insoluble ligand which offers binding sites to form mixed complexes.

In Figure 16, general structures for the Fe^3+^ based crosslinks between alginate and cellulose with involvement of carboxylic groups at C-6 or at the former reducing end of the cellulose are proposed.

In a combination of Ca^2+^ and Fe^3+^ complexation, the aggregation of alginate coated hematite nanoparticles could be controlled by addition of Mg^2+^ or Ca^2+^ ions. These reduced repulsion between the negatively charged nanoparticles and led to ionic crosslinks during aggregation [118]. For Ca^2+^ ions the formation of an extended Ca^2+^-alginate network surrounding the alginate coated hematite particles has also been reported by the authors. Thus, in this case, formation of aggregates is based on two principles. Bridging between two alginate coated hematite particles by calcium complexation and in a second phase, formation of calcium–alginate gels, which form gel–nanoparticle clusters. Resembling, ionic crosslinking of carboxylated cellulose nanofibrils could be observed by a number of bivalent and trivalent ions e.g., Fe^3+^, Al^3+^, Cu^2+^, Zn^2+^, and Ca^2+^ [119]. The complex formation with involvement of the carboxylate groups leads to formation of a hydrogel network. The interfibrillar carboxylate–cation interactions determine the final structure and properties of the gel formed.

Formation of iron (III)–polysaccharide complexes was also demonstrated as a method for the formation of anticoagulant coatings for medical products. In a layer-by-layer approach, combining FeCl_3_ layers of dextran sulfate and heparin, coatings with high hemocompatibility could be formed on nitinol sheets [120]. As mechanism for the layer formation the presence of iron complexes with involvement of anionic groups present in the polysaccharide, e.g., sulfate or carboxylate groups are supposed. The anticoagulant activity of the coating is explained by an outermost layer of bound heparin or dextran sulfate.

Similarly, cellulose films were modified with Ca^2+^ or Fe^3+^ cross-linked alginate layers to improve biocompatibility. In comparison to Ca^2+^ cross-linked alginate, the Fe^3+^ cross-linked layers showed higher adsorption of extracellular proteins and, thus, improved surface properties for cell adhesion and proliferation. This is of value in the modification of surfaces for implant material [121].

## 6. Ionic Interaction in Non- and Mixed-Polysaccharide Polymers

Multivalent ion crosslinking and dissolution through ion complexation is present in a multitude of systems. These mechanisms are not limited to hydroxylic or carboxylic functional groups but extend to amine, amide, imidazole [122], and hydroxyproline [123] groups, among others. Complexation is present in both polysaccharide and protein systems. These associating mechanisms are important to biological mechanisms. Glycoproteins, compounds of covalently bond oligosaccharides to protein chains, are important examples of mixed polysaccharide–protein systems [124,125]. These are known to show multivalent ion complexation [126,127,128]. Glycoprotein association is assessed by a relevant model. On the basis of a selected cross-linking and solubilizing mechanism of polyamide 6,6 and protein structures ionic cross-linking is extended to the latter. The following section is therefore added to:expand multivalent ion complexing and interaction modelsrefer to multivalent ion protein interaction and glycoproteins

### 6.1. Polyamide in CaCl_2_/Ethanol/Water Systems as the Model Compound for Polyamino Acid Structures

Polyamides, more specifically polyamide 6,6, can be considered to be synthetic polyamino acids. Natural polyamino acid structures occur for example in silk and wool. Polyamide 6,6 is inert to most common organic solvents and to alkali solutions, but is sensitive to acids such as sulfuric acid. It can be dissolved in concentrated formic acid, phenol, and alcoholic calcium chloride (CaCl_2_) solutions. The latter by formation of a Lewis acid–base complex.

The concept of Lewis acid–base complex formation between the polymer and Lewis acids can be applied to rigid chain and ladder polymers and to polyamides [129,130]. In general, Lewis acid–base complexation occurs between an electron donor (Lewis base) and an electron acceptor (Lewis acid). The formed bonds are called coordinate covalent bonds or dative covalent bonds. Compared to the covalent bonds the electrons are from the same atom.

Lewis acids (multivalent ions) mentioned in context with polyamide are for example GaCl_3_, AlCl_3_, CaCl_2_, BF_3_, BCl_3_, and LiCl, where GaCl_3_ and AlCl_3_ are considered to be strong Lewis acids and CaCl_2_ and LiCl weak Lewis acids. In polyamide the N–H and the C=O can act as electron donors, and thus be a complexation site for the Lewis acid. The nitrogen has a 2p_z_^2^ “lone pair” and the oxygen has two sp^2^ “lone pairs”. However, the 2p_z_ orbitals of O, C, and N in the planar amide group are overlapping, this causes the partial double bond character of the amide linkage and reduces the electron density of the nitrogen atom. This results in the formation of a coordinate bond between the carbonyl oxygen and the Lewis acid (see Figure 17) [131].

Nevertheless, these two sites are also the hydrogen bond donor and acceptor sites in the polyamide. Therefore, the hydrogen bond and the Lewis acid–base complex are competing. Complexation with polyamide suppresses the hydrogen bonding between the polymer chains, as sketched in Figure 18 for polyamide 6,6 and GaCl_3_ [129,130,131].

In most cases, the Lewis acids are dissolved in low molecular weight alcohols or in the case of GaCl_3_, nitromethane was used [129]. A more detailed dissolution mechanism of polyamide 6,6 in a CaCl_2_/methanol solution was suggested by Sun [132]. After adding polyamide to the solution, the Lewis acid–base complex approaches the polymer because of intermolecular interactions between the two groups, the O–H of the alcohol and the C=O of the polyamide. The affinity of the calcium ion is higher towards the oxygen of the carbonyl group than to the hydroxyl group, thus a transfer of the calcium ion occurs from the alcohol to the polyamide (Figure 19) [132].

However, the complexation of polyamide and Ca^2+^ ions is only happening in alcoholic solutions. Calcium chloride dissolved in water is surrounded by a solvation shell, thus it is thermodynamically very stable. By the addition of ethanol, the solvation shell is disturbed and a complex formation between the calcium ion and polyamide is thermodynamically preferred. The amount of water is playing an important role on the kinetics of the dissolution [133].

The dissolution is based on two different transport processes: diffusion and disentanglement. In solutions composed of calcium chloride and alcohol without water, these two processes are fast. The solvent diffuses into the polymer and breaks the hydrogen bonds between the polymer chains. At the same time, the polymer chains are disentangled and the polymer dissolves. The addition of water is believed to increase the diffusion rate into the polymer but the polymer chains, complexed with calcium ions, are not disentangling at the same rate, therefore a gel-like layer of complexed polyamide 6,6 emerges.

Different observations were made by Liu et al. [134] for polyamide 6 complexed with CaCl_2_. They found a red shift of the FTIR N–H vibration band to a lower frequency of 3245 cm^−1^, which has also been reported by Wu et al. [135] in polyamide 6,6 lithium salt complexes. They suggest a coordinating model, which includes the formation of a six-member ring between the salt and the polyamide chains (see Figure 20). In this ring, coordinate and hydrogen bonds coexist. This means the hydrogen bonds are not severed but are stronger than in the non-complexed polyamide [134,135].

Furthermore, the thermal properties of polyamide complexed with a Lewis acid are different compared to pristine polyamide [129]. This is caused by the absence of the hydrogen bonds. No melting point was observed, which indicates the lack of crystallinity of the complexed polyamide [129,130]. Polyamide 6,6 complexed with GaCl_3_ showed a T_g_ of –32 °C and no melting or crystallization [131,136].

Thus, complexation of polyamide with Lewis acids is also used to increase the draw ratio of polyamide fibers [137,138,139,140,141]. The salt, e.g., calcium chloride, is added to the melt and PA6_x_(CaCl_2_)_y_ fibers are melt-spun [140] or the polyamide, dissolved in formic acid and calcium chloride, is added to the solution and the fibers are prepared by the gel-spinning method [139].

The complexation of different salts (Lewis acids) with polyamide can be used as a model for the complexation of salts with proteins. By the complexation, the solubility and thermal and chemical properties of the polymer are changed.

### 6.2. Ion-Rich Hydrate Shells in Protein Fiber Dissolution

Silk fibroin consists of mostly linear protein chains, which in solid state are arranged in the β-pleated-sheet structure. Strong hydrogen bond interactions between neighbouring chains are present in the solid state. Due to the absence of covalent bonds, dissolution of fibroin can be achieved by a number of concentrated aqueous salt solutions, among them CaCl_2_/water/ethanol [12], CaNO_3_/methanol/water [142], LiBr [143], and NaSCN [144].

In solution considerable protein-protein associations between the fibroin chains lead to a strong tendency to aggregate and to form micelles [145].

FTIR analysis and potentiometric titration of the fibroin solution in concentrated CaCl_2_/water/ethanol allow to distinguish between complex formation and less specific interactions between the solvent and fibroin [146]. The common model assumes that dissolution leads to formation of an ion-rich hydration layer and interaction of calcium ions with charged and highly polar groups present in fibroin. Formation of well-defined Ca^2+^-complexes is less probable. This is supported by the low formation constants for Ca^2+^-complexes with the major fibroin constituents, e.g., glycine, alanine, tyrosine, and serine [147]. Similar to the D-gluconate complexes, amino acids with carboxylic side groups or basic amino groups could form more stable Ca^2+^ complexes however the share of these amino acids is not sufficient to explain fibroin dissolution in CaCl_2_/water/ethanol solution [148].

Based on similarities to the calcium complex with ethylenediaminetetraacetic (EDTA) acid, Figure 21 shows a model for a possible ion-interaction between Ca^2+^ ions and fibroin [149]. Involvement of NH-groups of the peptide chain contributes to the formation of five-membered chelate structures. For dissolution the complexation with the peptide backbone should be sufficient. Complexes of two peptide chains are expected to be present in solution.

In a similar manner, wool keratin can be dissolved using a concentrated solution of CaCl_2_/water/ethanol, however, addition of a suited reducing agent, e.g., thioglycolate, is required to open the covalent disulfide bonds between the keratin chains [150].

Weak complexes formed in the hydrate shell of an ion-rich solvent, thus can lead to dissolution of proteinaceous material in concentrated CaCl_2_/water/ethanol solutions. The regeneration of the chemically unmodified protein then is possible by dilution with non-solvents or application of dialysis. Such processes are highly interesting for shaping of protein based biomaterials in medical applications, e.g., scaffolds and implants.

### 6.3. Metal Complexes in Protein Fibers Forming Ionic

While concentrated solutions containing calcium chloride lead to dissolution of protein material via hydrate shell interactions, also stable and defined complexes can be formed between metal ions and a protein structure.

Soluble keratin hydrolysates were shown to form stable complexes with Fe^3+^ ions and with Cu^2+^ ions. Experiments with use of model peptides e.g., poly-l-lysine demonstrated the ability of these molecules to form stable metal complexes [151]. Formation of metal complexes with feather keratin could be obtained with use of glycine metal complexes (e.g., Zn^2+^, Cu^2+^, Mn^2+^, and Ni^2+^), which exhibits the ability of proteinaceous structures to act as solid ligands for bivalent metal ions [152] (Figure 22). Complex formation of metal ions with wool also was demonstrated with sorption experiments and polarographic analysis of the equilibrium concentrations using different metal ions (e.g., Cu^2+^, Pb^2+^, and Cd^2+^) and different processed wool samples [153].

Formation of larger assemblies of silk fibroin connected via divalent cations has been described to appear with silk fibroin [11,154]. Copper complexes, which crosslink two peptide chains through formation of a biuret type complex have been observed when silk fibroin has been treated with alkaline Cu^2+^ complex solutions, e.g., the cuprammonium complex [155] (Figure 23). The formation of Cu^2+^ complexes with silk fibroin was found to be dependent on the structure of the fibroin, e.g., fibroin in random coil structure forms Cu^2+^-chelate complexes more readily than fibroin present in antiparallel β-structure [144].

Similarly, the formation of a Ni^2+^ chelate complex could be achieved at pH > 10, while aqueous fibroin solutions gelated in presence of Ni^2+^ at pH < 9 [156].

## 7. Synopsis

A model of multivalent ion complexation from monosaccharides to polysaccharides was developed on the basis of selected studies. The final section expands the model to polyamino acids as model components for the ion-bridging in glycoproteins. In these considerations, the similarity of the behavior is demonstrated. The interactions derive from the ions species present as the polymers’ functional groups. Solvent systems and competitive bonds influence the ion complexation of the biopolymer, among others. The functional groups in a polysaccharide molecule determine the physical–chemical basis, the strength of interaction and possible complex formation with a certain ion. The interaction with ions present in solution, thus governs effects such as swelling, solubilisation, formation of a distinct conformation in the dissolved state (e.g., random coil, helical, or linear), as well as coagulation and precipitation.

## Figures and Tables

**Figure 1 molecules-25-01840-f001:**
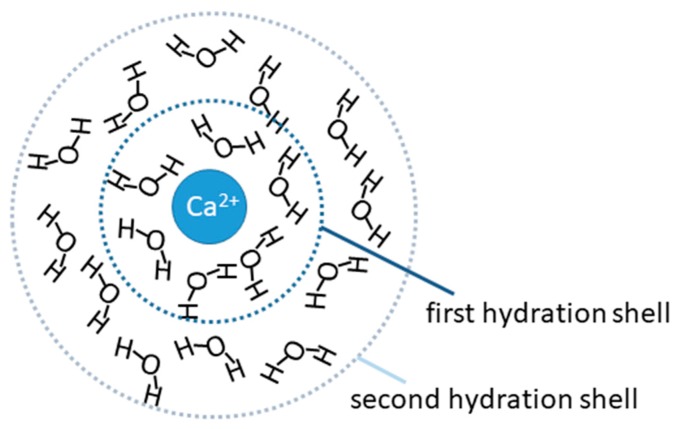
A 2D-representation of the first and second hydrate shell of a dissolved calcium ion.

**Figure 2 molecules-25-01840-f002:**
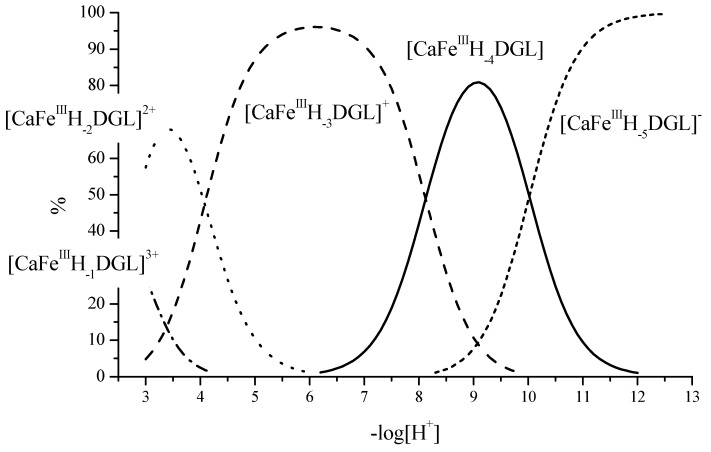
Representative example for complex species distribution calculated for the system Ca^2+^Fe^III^DGL (calcium-iron-d-gluconate) with a molar ratio 1:1:1 c(Ca^2+^) = 0.01 mol dm^−3^, c(Fe^3+^) = 0.01 mol dm^−3^, c(DGL) = 0.01 mol dm^−3^, (ionic strength µ(KNO_3_) = 0.1 mol dm^−3^, T = 20.0 ± 0.1 °C) and corresponding chemical reactions for complex formation.

**Figure 3 molecules-25-01840-f003:**
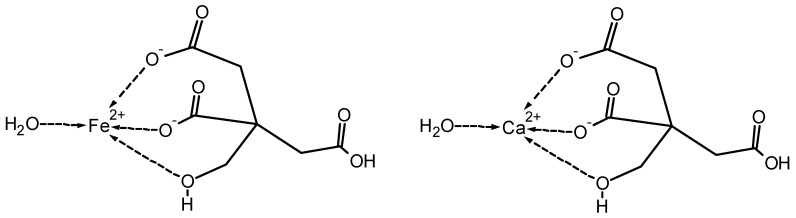
Representative structure of a Fe^2+^– and Ca^2+^–citrate complexes at neutral pH conditions (according to [21]).

**Figure 4 molecules-25-01840-f004:**
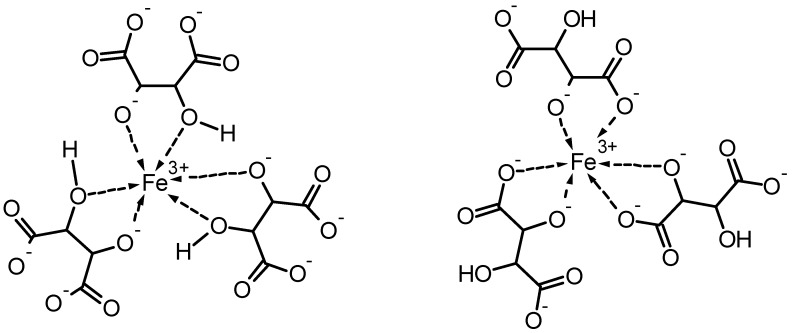
Proposed structure of the [(C_4_H_3_O_6_)_3_Fe]Na_6_ complex [26] (left) and alternative proposal in analogy to the iron–gluconate complexes with involvement of carboxylic groups (right).

**Figure 5 molecules-25-01840-f005:**
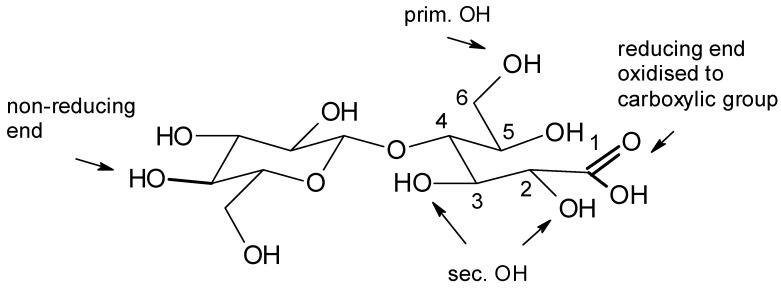
Functional groups present in processed cellulose.

**Figure 6 molecules-25-01840-f006:**
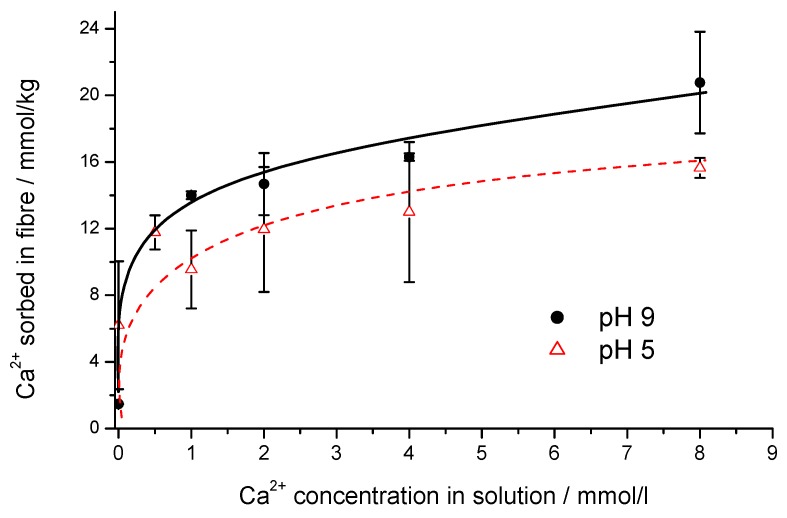
Binding of Ca^2+^ ions in the cellulose structure of a lyocell type fiber as function of Ca^2+^ concentration in solution at pH 9 and pH 5 [31].

**Figure 7 molecules-25-01840-f007:**
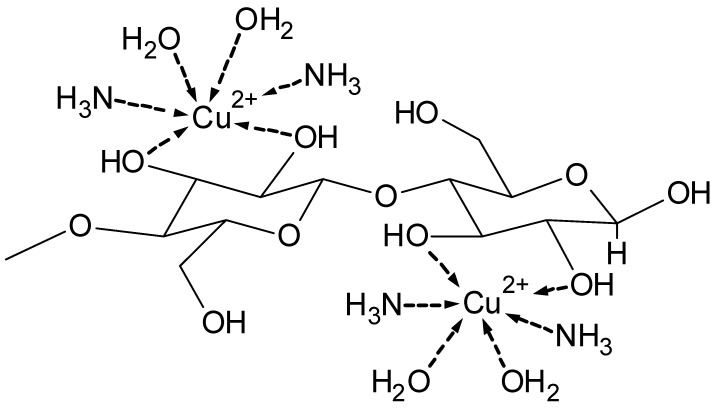
Complex between the cuprammonium system and cellulose [36].

**Figure 8 molecules-25-01840-f008:**
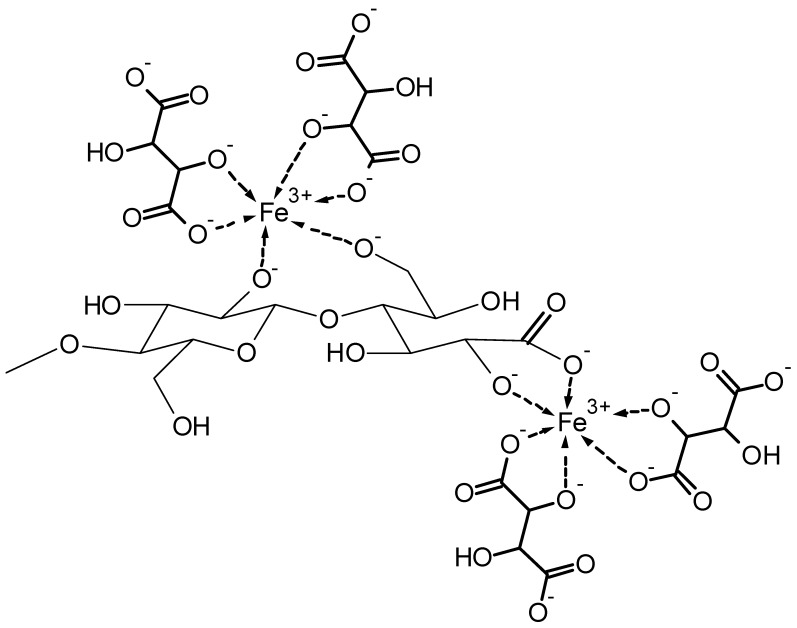
Proposed structure of the complex between the alkaline iron(III)–tartrate complex and cellulose.

**Figure 9 molecules-25-01840-f009:**
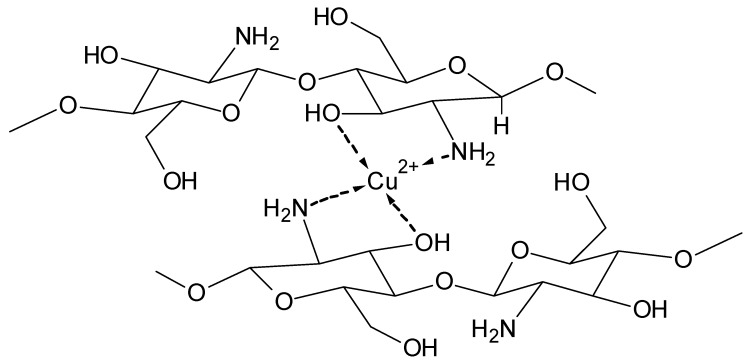
Proposed structure of metal complexes with chitosan [38].

**Figure 10 molecules-25-01840-f010:**
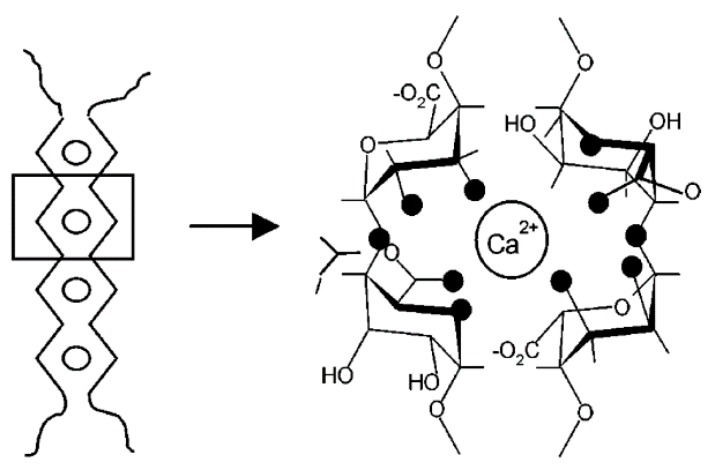
Possible calcium coordination of the “egg box model” as present in adjacent guluronate zones in alginates. Oxygen atoms participating in the coordination are darkly marked. (Reprinted with permission from Braccini, I. and Pérez, S. Molecular Basis of Ca^2+^-Induced Gelation in Alginates and Pectins: The Egg-Box Model Revisited. Biomacromolecules 2, 1089–1096 (2001). Copyright 2001 American Chemical Society).

**Figure 11 molecules-25-01840-f011:**
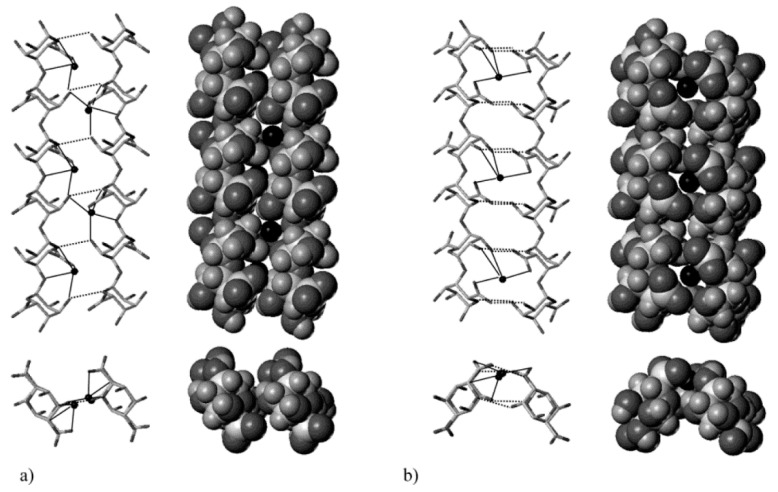
Representations (stick and van der Waals structures) of the best [chain–Ca^2+^–chain] associations of 2-fold guluronate chains: (**a**) parallel arrangement and (**b**) antiparallel arrangement. Positions of calcium ions have been reoptimized with respect to the dimer structures. Dark circles represent calcium ions. Key: (s) calcium coordination; (- - -) hydrogen bonds. (Reprinted with permission from Braccini, I. and Pérez, S. Molecular Basis of Ca^2+^-Induced Gelation in Alginates and Pectins: The Egg-Box Model Revisited. Biomacromolecules 2, 1089–1096 (2001). Copyright 2001 American Chemical Society).

**Figure 12 molecules-25-01840-f012:**
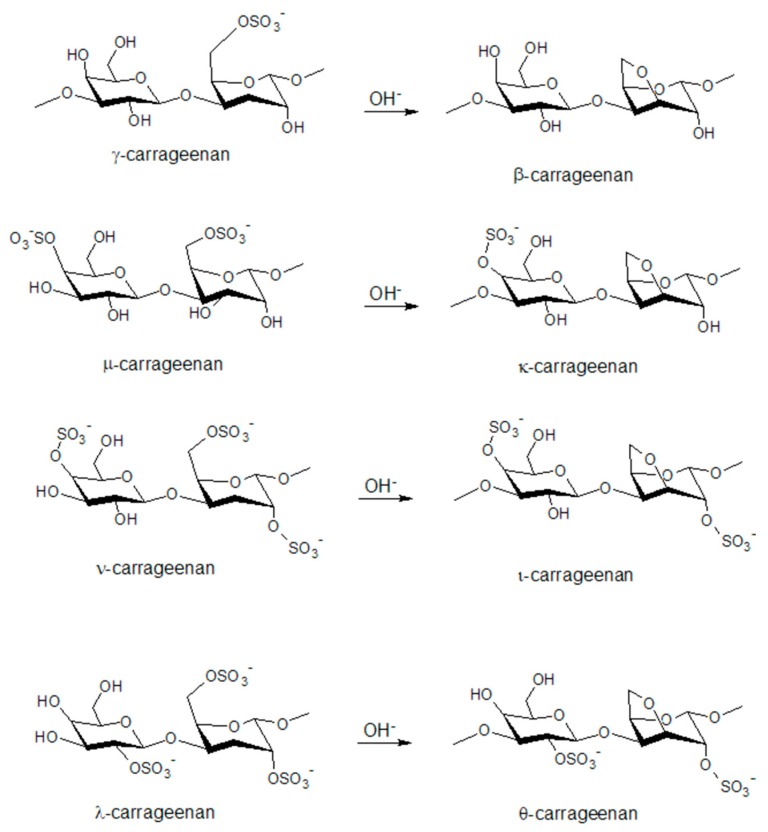
Chemical structures of disaccharide structures of the carrageenan types, as their possible interconversion.

**Figure 13 molecules-25-01840-f013:**
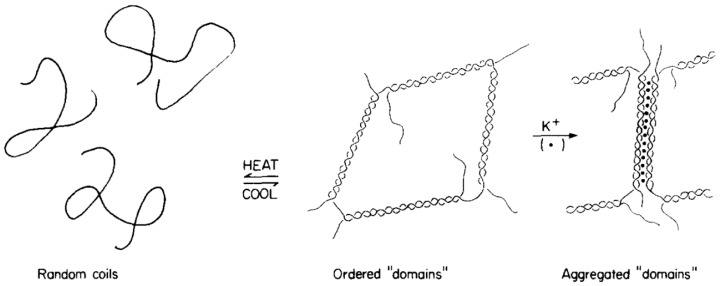
Domain model for carrageenan gelation (Reprinted from Morris, E. R., Rees, D. A. and Robinson, G., Cation-specific aggregation of carrageenan helices: Domain model of polymer gel structure, Journal of Molecular Biology 138, 349–362 (1980). Copyright (1980), with permission from Elsevier).

**Figure 14 molecules-25-01840-f014:**
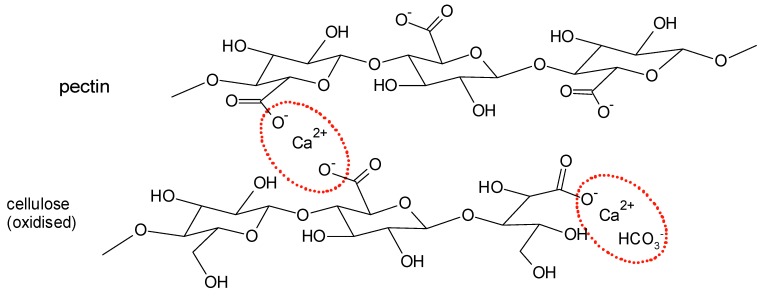
Proposed model structure for sorption of pectin on Ca^2+^-containing cellulose.

**Figure 15 molecules-25-01840-f015:**
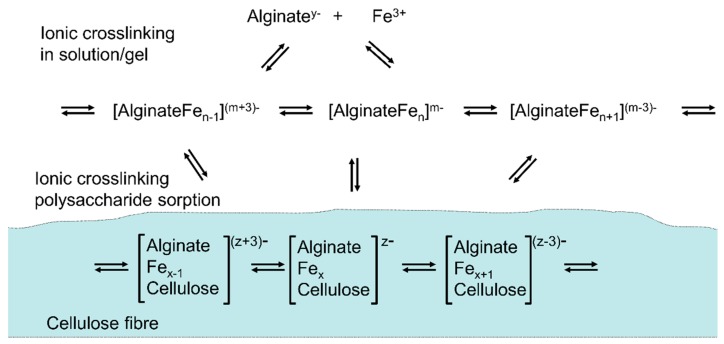
Sorption of polysaccharides (alginate–iron complexes) on cellulose through ionic crosslinking.

**Figure 16 molecules-25-01840-f016:**
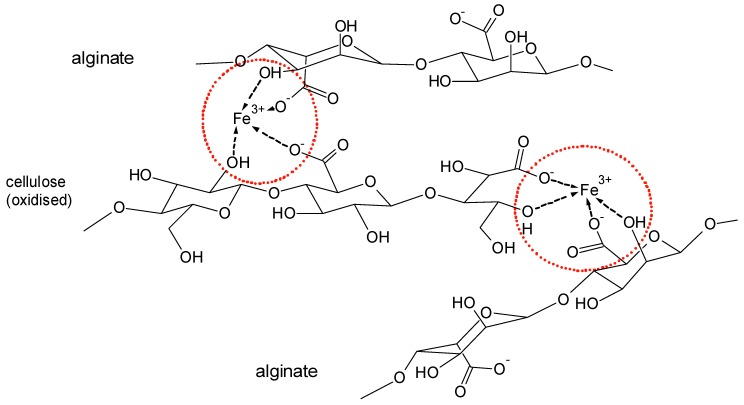
Proposed model structure for sorption of alginate on Fe^3+^-containing cellulose.

**Figure 17 molecules-25-01840-f017:**
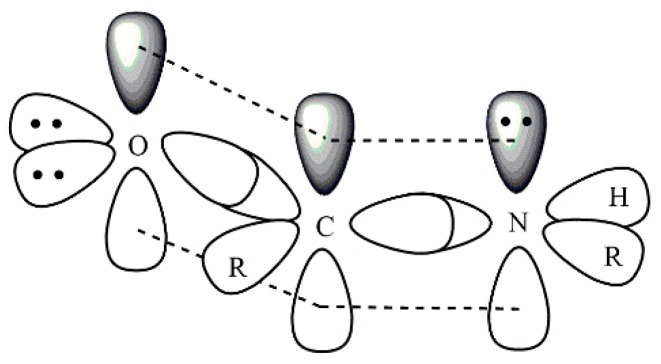
Orbitals of the amide group.

**Figure 18 molecules-25-01840-f018:**
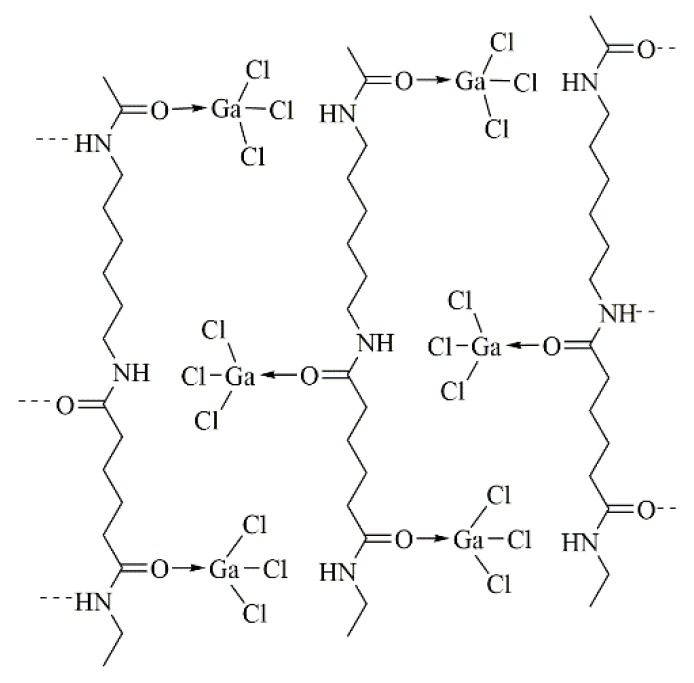
Polyamide 6,6 complexed with gallium chloride. Suggested structure by Roberts et al. [129,130].

**Figure 19 molecules-25-01840-f019:**
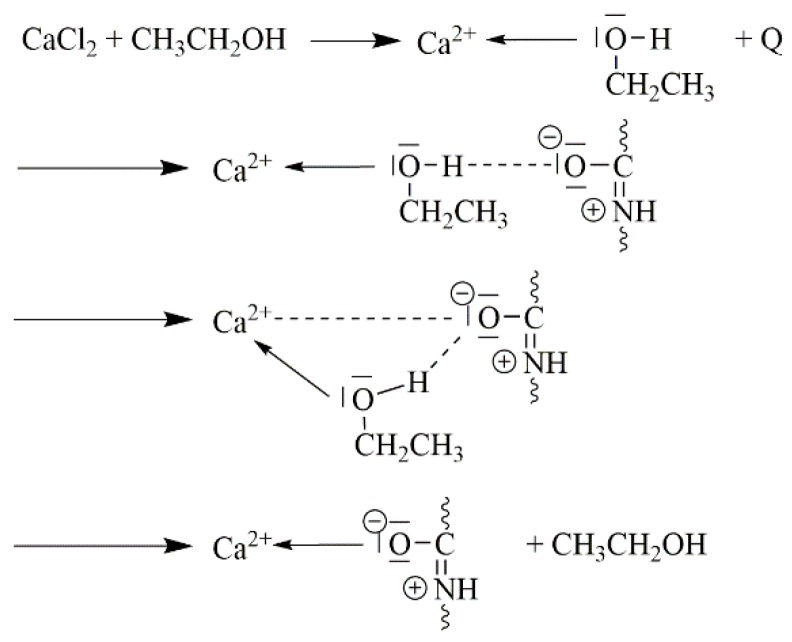
Suggested complexation mechanism of polyamide 6,6 and CaCl_2_/MeOH [132].

**Figure 20 molecules-25-01840-f020:**
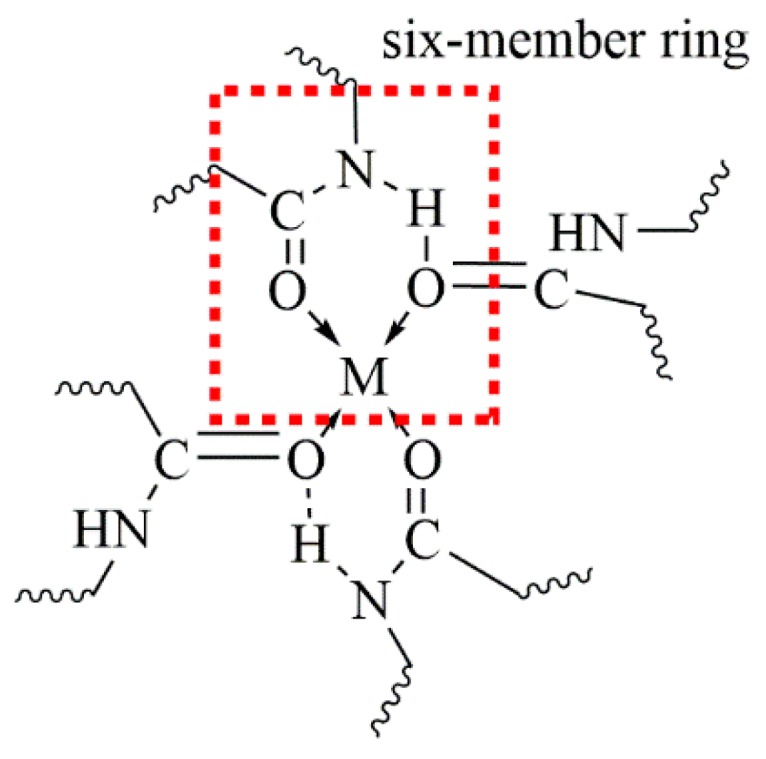
Suggested coordination model of polyamide and a Lewis acid (M) (Redrawn from [135]).

**Figure 21 molecules-25-01840-f021:**
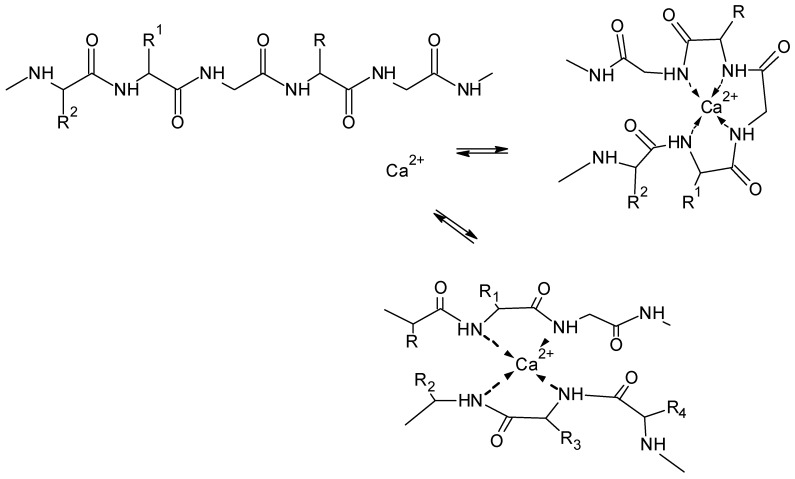
Schematic presentation of Ca^2+^ interaction with fibroin in CaCl_2_/water/ethanol solution.

**Figure 22 molecules-25-01840-f022:**
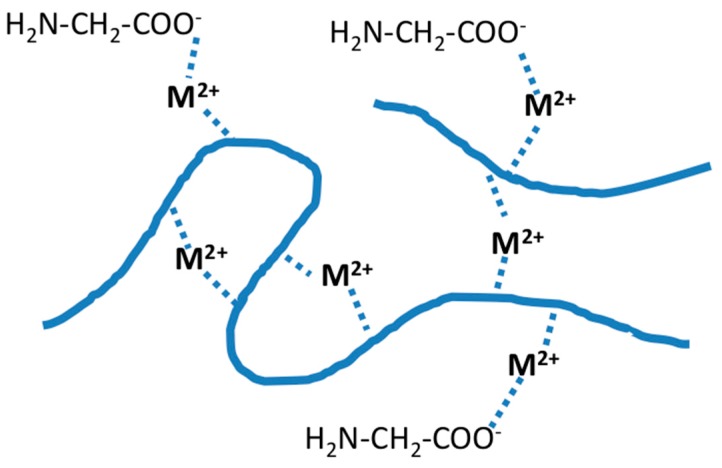
Formation of glycine metal complexes and metal complex crosslinks with feather keratin (M = Zn^2+^, Cu^2+^, Mn^2+^, and Ni^2+^) (based on [152]).

**Figure 23 molecules-25-01840-f023:**
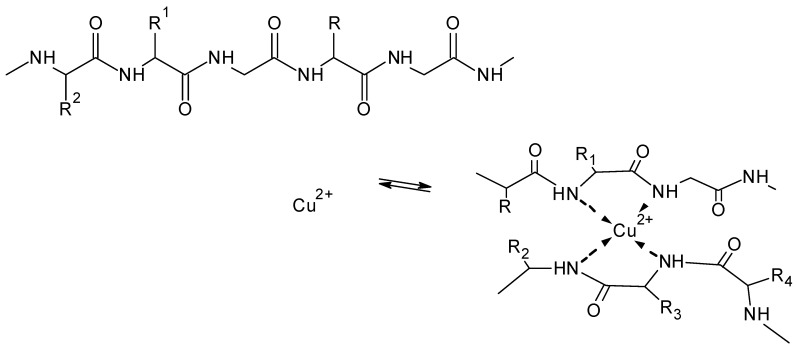
Formation of copper complexes with protein chains as basis for ionic crosslinking.

**Table 1 molecules-25-01840-t001:** Sulfate ester and 3,6-AG content of different carrageenan types. Data estimates from * [39,71] and ^+^ [72].

Carrageenan Type	Sulfate Ester Estimate (%)	3,6 AG Estimate (%)
β ^+^	0	26
Fur *	16–20	28–30
ι	32	26
κ	22	33
λ	37	~0

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
