# Peer review of "Multivalent Ions as Reactive Crosslinkers for Biopolymers—A Review"

_molecules, 2020, doi:10.3390/molecules25081840_

Round 1

Reviewer 1 Report

In the present manuscript, the authors review examples of ionic crosslinking of biopolymers applicable to the surface of textiles.

It is a subject of potential interest to the journal’s readership, and from the scientific point of view, the manuscript is adequate for publishing but before publication, the following aspects should be considered:

1) although the manuscript is generally written with minor grammar and syntax issues, thorough revision is required to correct several examples of poor phrase construction (for example lines 755-756).

Minor typos were detected (for example line 751).

Also, in some parts, the topics are organized in a confusing manner (for example, in lines 95-117, overall confusing and with no references cited) and with frequent repetition of the same ideas (for example lines 79-80 and 85-86).

The writing and formatting style should be standardized because the existence of several styles of writing by the different authors is notorious.

2) the introduction resembles an abstract, and it should be improved.

3) in line 92, the figure number (Fig 29.2) should be corrected to Fig 2.

4) In References, some formatting is required:

- references 57, 85, 102 and 103 refer to the same book, with different citation styles.

- references 128, 140, 141, 143 are incomplete.

Reviewer 2 Report

Wurm et al present their manuscript entitled “Multivalent Ions as Reactive Crosslinkers for Biopolymers on Fibre Surfaces and Interfaces”. In it, the authors describe essentially the hydrate shell of ions in water, the complexation with polymeric substances (both “sugars” and natural/synthetic aminoacid-based), including crosslinking reactions. In the view of this reviewer, this is a scientifically sound document – to no small account thanks to good English – thoroughly describing the physicochemical phenomena involved in the above mentioned reactions, generally well supported by references. Several materials are given as examples. This is important for readers working with biopolymers who need to understand ion-ion, ion-polymer and polymer-polymer interactions in hydrated conditions, and established structure-function relationships.

The manuscript, as received and despite its merit, requires significant attention to improve the overall presentation and to better describe the authors' scope. The authors are encouraged to address or to justify the following comments:

  1. The paper has at times is not sufficiently self-explanatory and these instances require attention. For example, the title can be misleading: it refers fiber surfaces and interfaces, but the text is not clear about the importance on focusing just in this type of systems. The introduction does not refer fibers or interfaces at all, though the abstract refers to textile fibers: the topic is only brought up in Chapter 5. Keywords also don’t reflect the importance of the fibers and interfaces in the presented revision. The authors should introduce an explanation at the start of the paper, or consider changing the title to be more general and broad.
  2. On page 2 (line 52), what do the authors mean by a “more or less” rapid ligand exchange?
  3. Section 2.1 mentions calcium, magnesium and iron ions in this order, going back again to calcium in the end. Please consider making the Ca2+ discussion in the same paragraph, as reading about other ions in the middle can be confusing.
  4. Still concerning the previous point, Ca2+ and Mg2+ descriptions are supported by references, but Fe3+ is not. Please verify if a citation is missing.
  5. Chapter 3 is entitled “Metal Ion Interactions…” whereas Chapter 4 is “Metal Ion Based Cross-linking…”. I my view, crosslinks are a type of interactions. The authors should consider changing Chapter 3 to for example to “Metal Ion Complexation…” or similar.
  6. On page 5 (lines 143-145) dairy processes are mentioned. In light of the intended focus on fibers and interfaces, the example seems somewhat irrelevant. How are the two related?
  7. Systems based on cellulose, alginate, carrageenans, xanthan, polyamides and silk are the materials receiving the greatest focus on this review. Again, the choice to review these materials in the intended context is not clear.
  8. On page 7, Equation 8 refers to cellulose as “Cell”. It should be clearer to the reader that this is not a biological cell they are referring to. In fact, cells and biological applications are mentioned in the review when mentioning alginate, so I advise using a different acronym for cellulose. The same applies to Equations 9 and 10 (page 18).
  9. Is Figure 8 an original picture? In the text reference 21 is cited for this figure, but not in the caption.
  10. On page 12 (lines 364 and 365), alginate gel beads are mentioned, but once again not the fibers (see also comments #1 and #6). Other examples are found in the text (e.g., in the case of pectins).
  11. Chapter 4.2 starts with the sentence: “A second representative of marine polysaccharides widely used are carrageenans”. Considering the scope of the review, do the authors mean marine polysaccharides that exhibit some kind of metal-ion based crosslinking? This comment is because one can think of other polysaccharides that better represent marine sources, such as chitosan.

Round 2

Reviewer 1 Report

The authors have made a comprehensive effort of improvement of the manuscript following the reviewer’s suggestions.

The manuscript is in adequate form to be considered for publication in this journal.

Reviewer 2 Report

The authors made a huge effort to address all the issues raised by this reviewer successfully and the manuscript in my view has been significantly improved.